# Optimizing Temporal and Spatial Efficiency for Chain-of-Thought Reasoning in Large Language Models

## Abstract

Chain-of-Thought (CoT) reasoning in Large Language Models (LLMs) achieves remarkable performance but suffers from significant computational overhead. CoT reasoning exhibits redundancy across two critical dimensions: temporal redundancy, where reasoning steps may be unnecessary, and spatial redundancy, where computations can be performed at reduced precision. While existing approaches require expensive dataset construction and model fine-tuning to improve reasoning efficiency, we propose *Temporal-Spatial Adaptive Reasoning (TSAR)*, a training-free framework that jointly exploits both redundancy dimensions through coordinated optimization. TSAR segments reasoning based on Dewey's reflective thinking model, employs progressive precision reduction that adapts to both reasoning phases and progress, and coordinates termination decisions through entropy-based confidence estimation. Our adaptive scheduler prevents precision-induced errors while enabling compound efficiency gains. Extensive evaluation on diverse reasoning tasks demonstrates up to $12.4\times$ speedup while maintaining the accuracy, establishing coordinated multi-dimensional redundancy exploitation as superior to conventional optimization strategies.

## 1 Introduction

Chain-of-Thought (CoT) reasoning has revolutionized how Large Language Models (LLMs) approach complex problem-solving, enabling systematic decomposition of intricate tasks through step-by-step reasoning (Wei et al., 2022; Parashar et al., 2025). However, this approach often produces long reasoning chains that are computationally expensive, as they require generating and processing a greater number of tokens across multiple intermediate steps, leading to higher inference latency, memory consumption, and energy demands (Sui et al., 2025; Feng et al., 2025). This poses significant challenges for practical deployment, particularly in resource-constrained environments.

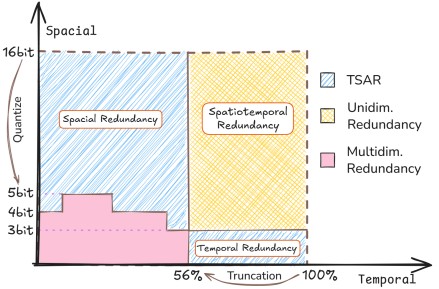

Figure 1: Reducing temporal and spatial redundancy is critical to optimize the efficiency of reasoning LLMs.

Existing approaches to improve CoT reasoning efficiency predominantly rely on training-dependent methods that require substantial computational resources and domain-specific data collection. Training-based methods construct specialized datasets and fine-tune models to reduce reasoning steps (Liu et al., 2024; Xia et al., 2025), but suffer from limited transferability across domains. Structural optimization approaches modify reasoning strategies through prompt engineering (Kang et al., 2024), but often sacrifice reasoning completeness for efficiency and require extensive manual optimization. Computational optimization techniques apply general LLM efficiency methods like uniform quantization (Zhang et al., 2025) or simple confidence-based early exit (Qiao et al., 2025) to reasoning tasks, but treat reasoning as generic text generation, missing reasoning-specific optimization opportunities and failing to exploit the multidimensional nature of redundancy in CoT reasoning.

The fundamental challenge in reducing redundancy for CoT reasoning lies in the multi-dimensional nature of inefficiency patterns. Specifically, modern reasoning LLMs exhibit redundancy across two critical dimensions (Figure 1): **temporal redundancy** where entire reasoning steps become unnecessary once sufficient understanding is achieved (Sui et al., 2025; Li et al., 2024), and **spatial redundancy**, where individual computations can be performed at reduced precision without significant quality degradation (Zhang et al., 2025). These redundancy dimensions are inherently correlated—they emerge from the same underlying reasoning dynamics and exhibit strong dependencies that current approaches systematically ignore. To effectively reduce these redundancies to optimize the inference efficiency of reasoning LLMs, there are several critical challenges that need to be addressed.

*Challenge 1: Redundancy Detection and Exploitation.* Identifying when reasoning steps become redundant requires a sophisticated understanding of reasoning progress and confidence dynamics. Simple heuristics like fixed confidence thresholds fail to capture the complex patterns of reasoning completion, while training-based approaches require expensive data collection and model modification. The challenge lies in developing training-free methods that can accurately detect redundancy patterns across both temporal and spatial dimensions without compromising reasoning quality.

*Challenge 2: Multi-Dimensional Coordination.* Temporal and spatial redundancy exhibit strong correlations that current independent optimization approaches fail to exploit. When precision reduction affects confidence estimation, termination mechanisms make suboptimal stopping decisions, leading to cascading errors. The challenge is developing coordinated optimization strategies that can exploit these correlations to achieve compound efficiency gains while preventing error propagation between dimensions.

*Challenge 3: Dynamic Adaptation to Reasoning Patterns.* CoT reasoning exhibits distinct computational phases with heterogeneous redundancy characteristics that static optimization cannot capture. E.g., problem formulation phases require high precision for accurate context establishment, while verification phases often tolerate aggressive quantization and early termination. The challenge lies in developing adaptive strategies that can dynamically adjust policies based on reasoning phase characteristics and temporal progress without requiring training or manual tuning.

To address these challenges, we propose *Temporal-Spatial Adaptive Reasoning (TSAR)*, a training-free framework that jointly optimizes temporal and spatial redundancy through coordinated multi-dimensional exploitation. TSAR operates by automatically segmenting reasoning sequences into distinct cognitive phases based on Dewey's reflective thinking model, enabling phase-specific optimization policies that adapt to heterogeneous computational requirements. The framework employs progressive precision reduction that dynamically adjusts quantization levels based on both reasoning phase characteristics and temporal progress within the reasoning sequence, moving beyond static uniform quantization to exploit the temporal dependencies inherent in step-by-step reasoning. TSAR coordinates termination decisions through entropy-based confidence estimation computed from thought transition patterns, preventing precision-induced confidence degradation from causing premature or delayed stopping.

The main contributions of this work can be summarized as follows,

- To the best of our knowledge, TSAR is the first training-free framework that jointly optimizes temporal and spatial redundancy through coordinated, phase-aware scheduling.

- We introduce dynamic precision allocation that adapts to both reasoning phase characteristics and temporal progress, extending beyond static uniform quantization to exploit reasoning-specific computational patterns.

- We establish unified scheduling where precision adaptation informs termination decisions, preventing cascading errors and enabling compound efficiency gains that exceed independent optimization approaches.

- Extensive evaluation on multiple reasoning tasks demonstrates that TSAR achieves up to $12.4\times$ speedup compared to the existing approach, establishing coordinated multi-dimensional redundancy exploitation as substantially superior to conventional efficiency optimization strategies.

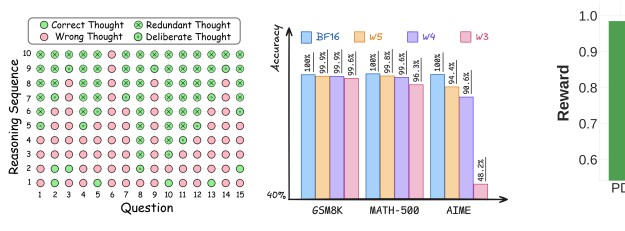

(a) Temporal redundancy.  (b) Spatial redundancy.

Figure 2: Temporal and spatial redundancy in reasoning LLMs.

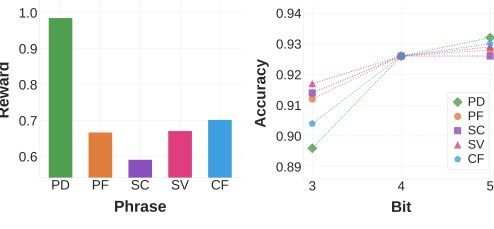

(a) Average reward.  (b) Quant. sensitivity.

Figure 3: Phase-aware reward and quantization sensitivity in reasoning LLMs.

## 2 MOTIVATIONS AND OBSERVATIONS

### 2.1 MULTI-DIMENSIONAL REDUNDANCY ANALYSIS

In order to quantitatively study the redundancy in large reasoning models, we conducted a study on the MATH-500 dataset (Hendrycks et al., 2021b). Our empirical analysis reveals redundancy patterns across two distinct dimensions that exhibit strong correlations, suggesting substantial potential for unified optimization strategies.

**Temporal redundancy** manifests when reasoning steps become unnecessary once sufficient understanding is achieved. For example, in the verification phases, reasoning often continues beyond the point where confidence stabilizes, with redundant verification steps providing minimal additional value while consuming substantial computational resources. Our analysis in Figure 2 (a) shows that 26.7% of reasoning steps could be eliminated without affecting final answer quality.

**Spatial redundancy** appears when computations can be performed at reduced precision without affecting reasoning quality. E.g., routine mathematical operations often do not require full precision, while complex analysis demands higher accuracy for numerical stability. Our statistical analysis in Figure 2 (b) reveals that this redundancy is substantial. For a majority of tasks, dramatically reducing precision from full BF16 down to 4-bits (w4), and in some cases even 3-bits (w3), incurs only a negligible drop in accuracy.

### 2.2 PHASE-BASED COMPUTATIONAL REQUIREMENTS

We hypothesize that CoT reasoning is not a monolithic computational process but rather consists of distinct phases with heterogeneous resource requirements. To test this, we conducted an experiment grounded in the principles of John Dewey's model of reflective thinking (Dewey, 1933).

Specifically, we split the thoughts generated by LLMs into these phases (with the method introduced in the Methodology Section) and analyze them using a large reward model. Specifically, we employ the reward model Qwen2.5-Math-PRM-7B (Yang et al., 2024) to statistically analyze the scores across these stages, systematically adjusting the precision of each stage to observe its impact on overall accuracy. As illustrated in Figure 3(a), we observe distinct reward magnitudes for different stages. Furthermore, Figure 3(b) reveals their varying sensitivity to quantization. For instance, the Problem Decomposition (PD) phase, which has the highest reward, is highly sensitive to precision, showing a steep decline in accuracy at lower bit-widths. In contrast, other phases like Solution Consolidation (SC) are far more robust to quantization, maintaining high accuracy even at 3-bit precision. This evidence supports our claim that different reasoning phases have distinct computational requirements.

These phase-dependent patterns reveal that optimal efficiency strategies must adapt dynamically rather than apply uniform policies. Static quantization fundamentally cannot capture these variations, leading to systematic resource misallocation where high-redundancy phases receive excessive computational resources while precision-critical phases may be under-provisioned.

Figure 4: Framework of TSAR, which effectively identifies different reasoning phases, dynamically adjusts their precision levels and timely enables early termination.

## 3 METHODOLOGY

In this section, we introduce the TSAR framework, depicted in Figure 4, orchestrating phase-aware precision scheduling and entropy-based termination to optimize temporal and spatial efficiency.

### 3.1 PROBLEM FORMULATION AND REDUNDANCY DEFINITIONS

We formalize the multi-dimensional redundancy optimization problem for CoT reasoning. Consider model $M_\theta$ generating reasoning sequence $\mathbf{s} = (s_1, s_2, \ldots, s_T)$ given input $\mathbf{x}$, where the $t$-th thought $s_t$ is computed using precision $p_t \in \mathcal{P} = \{2, 3, 4, \cdots\}$ bits with computational cost $c(p_t)$.

**Temporal Redundancy** $R_T(t)$ quantifies the unnecessary computational cost from continuing reasoning beyond sufficient understanding:

$$RT(t) = c(p_t) \cdot \mathbf{1}[t > t^*] \quad \text{where } t^* = \min\{t' \mid \mathcal{Q}(s_{1:t'}) \geq \mathcal{Q}(s_{1:T}) - \epsilon\} \tag{1}$$

where $\mathcal{Q}(\cdot)$ measures reasoning quality and $t^*$ is the optimal reasoning depth and $\epsilon$ represents acceptable quality tolerance.

**Spatial Redundancy** $R_S(t)$ quantifies computational redundancy from precision reduction without quality loss:

$$R_S(t) = c(p_t) - c(p_t^*) \quad \text{where } p_t^* = \min\{p \in \mathcal{P} : \mathcal{Q}(s_t, p) \geq \mathcal{Q}(s_t, p_{max}) - \delta\} \tag{2}$$

where $\delta$ represents spatial quality tolerance.

### 3.2 REASONING PHASE CLASSIFICATION

Inspired by our observation of phase-dependent computational needs, we structure our adaptive framework around a cognitive model. John Dewey's reflective thinking model, as outlined in his foundational work How We Think (Dewey, 1933), provides a cognitive framework for thoughtful inquiry and problem-solving. It describes a sequential process involving five key phases: (1) the recognition of a felt difficulty or problem, (2) the location and intellectualization of the problem (defining it clearly), (3) the suggestion of possible solutions or hypotheses, (4) the development of these suggestions through reasoning and deduction, and (5) the testing of hypotheses through observation, experimentation, or verification, leading to acceptance or rejection.

Building upon this cognitive science foundation, we adapt Dewey's model to the context of LLM reasoning, distilling it into five distinct phases that align with its core principles while emphasizing heterogeneous computational characteristics. These phases enable targeted optimizations that traditional static approaches cannot exploit effectively:

- Problem Definition (PD), which corresponds to Dewey's problem recognition (phases 1), focusing on initial context establishment;
- Problem Formulation (PF), which extends the intellectualization by formalizing problem structures, constraints, and representations (phases 2);
- Solution Computation (SC), akin to hypothesis suggestion and initial development (phase 3), involving generative reasoning and derivation;

- Solution Verification (SV), reflecting further reasoning, deduction, and testing (phase 4), emphasizing validation and error checking;
- Conclusion Formation (CF), mirroring final verification and acceptance (phase 5), synthesizing results into a coherent outcome.

We segment reasoning sequences at thought transition points identified by linguistic indicators:

$$\mathcal{S}(s_t) = \mathbf{1}[\exists k \in \mathcal{K}_{split} : k \in s_t] \tag{3}$$

where $\mathcal{K}_{split} = \{"wait", "Wait", "Alternatively", \cdots\}$.

In order to ensure operational efficiency, we adopted a keyword-based classification method in the reasoning stage. Specifically, each segment is classified using enhanced phase detection:

$$\Phi(segment) = \arg\max_{\psi \in \Psi} \sum_{k \in \mathcal{K}_\psi} f_k(segment) \tag{4}$$

where $\Psi = \{PD, PF, SC, SV, CF\}$, and $f_k$ is the keyword counting function of keyword $k$.

### 3.3 Phase-Aware Progressive Quantization

Traditional quantization methods require separate model copies for each precision level, creating prohibitive memory overhead for dynamic switching. Any-Precision maintains a single model that can operate at multiple precisions through nested quantization (Park et al., 2024), enabling seamless precision adaptation essential for reasoning optimization. This capability is particularly valuable for reasoning tasks where precision requirements can change within a single reasoning sequence.

Our dynamic precision allocation adapts based on phase characteristics, reasoning progress, and confidence. The scheduler is guided by a principled linear model, which serves as a computationally efficient, first-order approximation of an ideal precision function. For a detailed justification, please see Appendix B. The model is defined as:

$$p_t = \max\left(p_{base}(\phi_t) - \alpha \cdot \frac{t}{T_{exp}} + \beta \cdot (1 - C(t)), 2\right) \tag{5}$$

where $p_{base}(\phi_t)$ provides phase-specific baseline precision. $T_{exp}$ is the expected number of reasoning thought and $\alpha \cdot \frac{t}{T_{exp}}$ implements progressive reduction aligned with reasoning advancement. $\beta \cdot (1 - C(t))$ increases precision when confidence is low. This formulation provides a robust and interpretable control strategy without the overhead of complex nonlinear models.

### 3.4 Coordinated Termination

Our termination strategy coordinates precision adaptation with stopping decisions through entropy-based confidence estimation computed from thought transition patterns. This coordination prevents precision-induced confidence degradation from causing suboptimal termination decisions.

For reasoning steps containing split tokens (in most cases also the important high-entropy tokens (Wang et al., 2025)) indicating reconsideration, we compute transition entropy as a measure of reasoning uncertainty:

$$H_t = -\sum_{v \in V_{split}} P(v|context_t) \log P(v|context_t) \tag{6}$$

where $V_{split}$ contains transition vocabulary indicative of reasoning uncertainty and $P(v|context_t)$ represents the probability of generating transition token $v$ given current reasoning context. Our unified confidence metric integrates phase-specific quality assessment with temporal confidence dynamics as:

$$C(t) = \omega_1 \cdot r_{\phi(s_t)} + \omega_2 \cdot \frac{H_{base} - H_t}{H_{base}} \tag{7}$$

where $r_{\phi(s_t)}$ represents phase-specific quality, and $\omega_i$ are weighting parameters. The coordinated termination decision integrates multiple criteria:

$$\mathcal{T}(t) = (C(t) \geq \tau_e) \wedge (t \geq t_{min}) \tag{8}$$

where $t_{min}$ enforces minimum number of thoughts and $\tau_e$ is the threshold. To implement natural termination, we employ confidence-based completion phrase injection. When termination criteria are satisfied, we append contextually appropriate completion indicators like "Okay, I think I have finished thinking" This approach maintains reasoning coherence while enabling efficient early exit.

## 4 EXPERIMENTS

### 4.1 EXPERIMENTAL SETUP

**Models and Datasets.** We conduct comprehensive evaluation using two model variants: *DeepSeek-R1-Distill-Qwen-7B* (Guo et al., 2025) and *Qwen-3-8B* (Yang et al., 2025), implementing our framework with the Any-Precision quantization algorithm (Park et al., 2024) to enable seamless dynamic precision switching without memory overhead. To demonstrate TSAR's applicability across different architectures. We evaluate performance on four challenging mathematical reasoning datasets that cover a wide spectrum of complexity: GSM8K (Cobbe et al., 2021), which consists of elementary-level word problems requiring sequential arithmetic reasoning; MATH-500, a high-difficulty subset of 500 competition-style problems derived from the MATH dataset; AIME-120, featuring problems from the American Invitational Mathematics Examination (2022–2025) that test advanced problem-solving techniques; and AMC-23, a selection of problems from the American Mathematics Competitions that assess deep conceptual understanding and strategic reasoning. In addition, we conducted evaluations on two general-purpose datasets: GPQA-Diamond-MC (Rein et al., 2024), a collection of 198 highly challenging questions from the Graduate-Level Google-Proof Q&A benchmark; and MMLU (Hendrycks et al., 2021a), a benchmark spanning numerous academic and professional domains, designed to assess models' general-purpose knowledge and reasoning capabilities.

**Baselines and Metrics.** We establish comprehensive baselines representing different optimization paradigms to evaluate TSAR's effectiveness across multiple dimensions.

*Static Quantization Baselines.* We implement uniform quantization strategies where all reasoning phases operate at fixed precision levels: 3-bit uniform quantization and 4-bit uniform quantization (Frantar et al., 2022). These baselines represent conventional static approaches that cannot adapt to dynamic reasoning requirements.

*Adaptive Quantization Baseline.* Following the Progressive Mixed-Precision Decoding (PMPD) framework (Chen et al., 2025a), we implement a naive scheduler that switches from high-precision to low-precision models. This approach provides temporal adaptation without reasoning awareness, serving as a direct comparison for our coordinated strategy.

*Early Termination Baseline.* We compare our approach with two training-free early stopping methods, S1 length control (Muennighoff et al., 2025) and NoThinking (Ma et al., 2025a).In the comparison, we align their generation budgets with ours to enable a fair evaluation of performance.

### 4.2 MAIN RESULTS

As presented in Table 1, TSAR demonstrates consistent and substantial superiority over all baselines across two distinct models and a wide spectrum of reasoning tasks. Our analysis is structured by task difficulty, revealing how TSAR's coordinated optimization excels at every level of complexity.

**Dominant Efficiency on Foundational Reasoning Tasks (GSM8K).** On elementary mathematical problems like GSM8K, where reasoning paths are relatively straightforward, TSAR delivers massive efficiency gains with virtually no accuracy trade-off. Across both models, it achieves a *5.7×-6.2× efficiency boost* while maintaining accuracy that is statistically indistinguishable from the full-precision baseline (e.g., 0.944 vs. 0.955 on Qwen-3). This performance highlights the efficacy of our entropy-based termination criterion, which accurately detects the point of confidence saturation and prunes the redundant reasoning chain. In contrast, while S1 length-control also reduces tokens, its reliance on full precision makes it spatially inefficient (only 1.1×-1.2× total gain).

Table 1: Model performance on various reasoning datasets. Efficiency is measured as a composite of temporal and spatial efficiency.

| Method | DeepSeek-R1-Distill-Qwen-7B | | | | Qwen-3-8B | | | |
|---|---|---|---|---|---|---|---|---|
| | Accuracy | Avg. Bit | Avg. Tokens | Efficiency | Accuracy | Avg. Bit | Avg. Tokens | Efficiency |
| Dataset | **GSM8K Dataset** | | | | | | | |
| Original | 0.923 | 16.00 | 1847.16 | 1.0× | 0.955 | 16.00 | 2335.76 | 1.0× |
| Uniform 4-bit | 0.922 | 4.00 | 1822.44 | 4.1× | 0.953 | 4.00 | 2368.36 | 3.9× |
| Uniform 3-bit | 0.919 | 3.00 | 1857.97 | 5.3× | 0.924 | 3.00 | 2459.24 | 5.1× |
| PMPD | 0.920 | 3.52 | 1835.73 | 4.6× | 0.938 | 3.53 | 2312.44 | 4.6× |
| NoThinking | 0.903 | 16.00 | 1984.76 | 0.9× | 0.942 | 16.00 | 2016.14 | 1.2× |
| S1 length-control | 0.920 | 16.00 | 1526.43 | 1.2× | 0.932 | 16.00 | 2162.08 | 1.1× |
| TSAR (Ours) | 0.923 | 3.51 | 1353.77 | **6.2×** | 0.944 | 3.54 | 1845.09 | **5.7×** |
| Dataset | **MATH-500 Dataset** | | | | | | | |
| Original | 0.930 | 16.00 | 4186.74 | 1.0× | 0.950 | 16.00 | 5469.73 | 1.0× |
| Uniform 4-bit | 0.926 | 4.00 | 4202.62 | 4.0× | 0.922 | 4.00 | 5397.44 | 4.1× |
| Uniform 3-bit | 0.896 | 3.00 | 4223.48 | 5.3× | 0.870 | 3.00 | 6140.33 | 4.8× |
| PMPD | 0.826 | 3.49 | 4161.34 | 4.6× | 0.924 | 3.56 | 5425.36 | 4.5× |
| NoThinking | 0.890 | 16.00 | 3309.78 | 1.3× | 0.926 | 16.00 | 4922.34 | 1.1× |
| S1 length-control | 0.908 | 16.00 | 2836.54 | 1.5× | 0.875 | 16.00 | 4727.32 | 1.2× |
| TSAR (Ours) | 0.922 | 3.61 | 2993.74 | **6.2×** | 0.928 | 3.63 | 4506.80 | **5.3×** |
| Dataset | **AIME-120 Dataset** | | | | | | | |
| Original | 0.450 | 16.00 | 14909.66 | 1.0× | 0.675 | 16.00 | 17242.02 | 1.0× |
| Uniform 4-bit | 0.408 | 4.00 | 14140.91 | 4.2× | 0.667 | 4.00 | 15061.76 | 4.6× |
| Uniform 3-bit | 0.217 | 3.00 | 12481.43 | 6.4× | 0.292 | 3.00 | 18522.59 | 5.0× |
| PMPD | 0.342 | 3.46 | 12188.29 | 5.7× | 0.583 | 3.53 | 18134.53 | 4.3× |
| NoThinking | 0.283 | 16.00 | 9267.87 | 1.6× | 0.375 | 16.00 | 14862.65 | 1.2× |
| S1 length-control | 0.317 | 16.00 | 10126.44 | 1.5× | 0.492 | 16.00 | 13653.46 | 1.3× |
| TSAR (Ours) | 0.383 | 3.47 | 9899.67 | **6.9×** | 0.633 | 3.31 | 12683.75 | **6.6×** |
| Dataset | **AMC-23 Dataset** | | | | | | | |
| Original | 0.900 | 16.00 | 6854.68 | 1.0× | 0.900 | 16.00 | 9756.05 | 1.0× |
| Uniform 4-bit | 0.900 | 4.00 | 6660.15 | 4.1× | 0.900 | 4.00 | 8560.50 | 4.6× |
| Uniform 3-bit | 0.825 | 3.00 | 7929.35 | 4.6× | 0.800 | 3.00 | 11231.74 | 4.6× |
| PMPD | 0.825 | 3.45 | 7342.91 | 4.3× | 0.825 | 3.45 | 8617.82 | 5.3× |
| NoThinking | 0.775 | 16.00 | 4856.43 | 1.4× | 0.850 | 16.00 | 8751.86 | 1.1× |
| S1 length-control | 0.850 | 16.00 | 5097.83 | 1.3× | 0.850 | 16.00 | 6423.78 | 1.5× |
| TSAR (Ours) | 0.875 | 3.64 | 4642.91 | **6.5×** | 0.875 | 3.43 | 6150.85 | **7.4×** |
| Dataset | **GPQA-Diamond-MC Dataset** | | | | | | | |
| Original | 0.525 | 16.00 | 10268.76 | 1.0× | 0.586 | 16.0 | 10229.74 | 1.0× |
| Uniform 4-bit | 0.520 | 4.00 | 8769.02 | 4.7× | 0.566 | 4.00 | 10170.12 | 4.0× |
| Uniform 3-bit | 0.409 | 3.00 | 8123.29 | 6.7× | 0.429 | 3.00 | 10503.03 | 5.2× |
| PMPD | 0.455 | 3.51 | 8245.86 | 5.7× | 0.530 | 3.55 | 10435.67 | 4.4× |
| NoThinking | 0.389 | 16.00 | 3895.11 | 2.7× | 0.505 | 16.00 | 8802.09 | 1.2× |
| S1 length-control | 0.449 | 16.00 | 4923.51 | 2.1× | 0.484 | 16.00 | 8531.45 | 1.2× |
| TSAR (Ours) | 0.520 | 3.53 | 3762.13 | **12.4×** | 0.545 | 3.55 | 8204.79 | **5.6×** |
| Dataset | **MMLU Dataset** | | | | | | | |
| Original | 0.634 | 16.00 | 1958.83 | 1.0× | 0.826 | 16.00 | 2252.57 | 1.0× |
| Uniform 4-bit | 0.633 | 4.00 | 1821.87 | 4.3× | 0.825 | 4.00 | 2089.12 | 4.3× |
| Uniform 3-bit | 0.592 | 3.00 | 2029.23 | 5.1× | 0.791 | 3.00 | 2124.62 | **5.7×** |
| PMPD | 0.603 | 3.54 | 1523.49 | 5.8× | 0.805 | 3.58 | 2203.87 | 4.6× |
| NoThinking | 0.571 | 16.00 | 1162.55 | 1.7× | 0.786 | 16.00 | 1857.23 | 1.2× |
| S1 length-control | 0.612 | 16.00 | 1547.64 | 1.3× | 0.812 | 16.00 | 2135.45 | 1.1× |
| TSAR (Ours) | 0.633 | 3.55 | 1258.71 | **7.0×** | 0.823 | 3.62 | 1790.82 | 5.6× |

Furthermore, the majority voting mechanism employed by the NoThinking method not only fails to achieve spatial efficiency but also yields inconsistent accuracy across different models. More importantly, uniform 3-bit quantization shows a noticeable accuracy drop (e.g., 0.924 on Qwen3-8B), proving that even simple tasks can be sensitive to naive, static compression. TSAR's dynamic approach avoids this pitfall, providing a near-optimal balance of temporal and spatial efficiency for this task category.

**Navigating the Trade-off in Complex Problems (MATH-500).**   As task complexity increases, the accuracy-efficiency trade-off becomes critical. On the challenging MATH-500 dataset, TSAR

proves its ability to navigate this trade-off effectively. It retains over *97% of the original accuracy* on both models (99.1% for DeepSeek-R1, 97.7% for Qwen-3) while delivering a $5.3\times$-$6.2\times$ efficiency gain. This success is directly attributable to our phase-aware scheduling. The framework allocates higher precision to critical initial phases, then progressively reduces it, preventing the kind of catastrophic errors seen in baselines. For instance, the PMPD scheduler, with its one-way switch from high to low precision, is too brittle for this complexity, causing a *severe 11.2% accuracy drop* on DeepSeek-R1. Under comparable token counts relative to TSAR while preserving their original precision, both S1 length-control and the NoThinking method demonstrate unstable accuracy at an efficiency ratio of $1.1\times$–$1.5\times$, significantly underperforming compared to our method. This demonstrates that as problems become harder, a simple temporal schedule is insufficient; a coordinated, context-aware strategy like TSAR's is required to succeed.

**Robustness Under Pressure in Contest-Level Scenarios (AIME-120, AMC-23).** The superiority of TSAR is most pronounced on high-difficulty contest benchmarks, where baseline methods often collapse. On AIME-120, TSAR preserves an impressive *85.1% (DeepSeek-R1) and 93.8% (Qwen-3) of original accuracy* with over $6\times$ efficiency. On AMC-23, TSAR also achieve an efficiency gain of $6.5\times$-$7.4\times$. PMPD's performance, however, plummets, demonstrating its inability to handle sophisticated reasoning under compression. These findings strongly validate that our coordinated, phase-aware optimization is not merely beneficial but *essential* for preserving reasoning capabilities in the most demanding scenarios.

**Broad Applicability on General-Purpose Reasoning (MMLU, GPQA-Diamond-MC).** To confirm that TSAR's benefits extend beyond mathematical domains, we evaluated it on the multi-domain MMLU benchmark. The results affirm its broad applicability. TSAR again achieves a *high efficiency gain (5.6$\times$-7.0$\times$)* while matching the original model's accuracy (e.g., 0.633 vs. 0.634 on DeepSeek-R1; 0.823 vs. 0.826 on Qwen-3). Similarly, on the challenging GPQA-Diamond-MC dataset, TSAR opens up a significant accuracy gap of up to *6.5% over PMPD, 7.1% over S1 and 13.1% over NoThinking* (on DeepSeek-R1). Our method achieves a significant improvement in both accuracy and efficiency, despite operating at comparable precision to PMPD and similar token counts to S1 length-control and the NoThinking. Although Uniform 3-bit quantization attains the best efficiency on the Qwen-3-8B model, its considerable loss in accuracy cannot be overlooked. This demonstrates that TSAR's principles of identifying and exploiting temporal-spatial redundancy are task-agnostic and highly effective for general-purpose reasoning as well.

## 4.3 ABLATION STUDY

We conducted systematic ablation studies to evaluate the impact of each TSAR component by progressively reducing its parameter values to 0 while keeping other components fixed. Using the MATH-500 benchmark, we measured performance through two key metrics: (1) per-question average token count (indicative of computational efficiency) and (2) dataset-wide accuracy (reflecting overall task performance).

Table 2: Ablation Study Results.

| $\omega_1$ | 0 | 0.1 | 0.2 | 0.3 | 0.4 | 0.5 |
|---|---|---|---|---|---|---|
| Accuracy | 0.904 | 0.904 | 0.906 | 0.910 | 0.916 | 0.924 |
| Avg. Tokens | 2651.23 | 2690.37 | 2715.69 | 2954.88 | 3242.13 | 3460.92 |
| $\omega_2$ | 0 | 0.1 | 0.2 | 0.3 | 0.4 | 0.5 |
| Accuracy | 0.934 | 0.926 | 0.920 | 0.914 | 0.908 | 0.906 |
| Avg. Tokens | 4232.07 | 3369.71 | 3028.56 | 2936.48 | 2869.32 | 2806.09 |
| $t_{min}$ | 0 | 1 | 2 | 3 | 4 | 5 |
| Accuracy | 0.914 | 0.922 | 0.920 | 0.922 | 0.926 | 0.932 |
| Avg. Tokens | 2762.35 | 2993.74 | 3243.45 | 3410.62 | 3784.10 | 3915.86 |

As detailed in Table 2, The study reveals three key results:

- *Ablation of phase quality weight:* By gradually reducing $\omega_1$ to 0, we observe that the accuracy first drops rapidly before stabilizing at 0.904, while the token count decreases sharply and then plateaus around 2,600 tokens.

- *Ablation of entropy shift weight:* Similarly, when reducing $\omega_2$ to 0, the reasoning sequence $s$ loses its constraints. Consequently, the truncation point of our method depends solely on textual phrasing, leading to a gradual increase in both token count and accuracy. This clearly demonstrates TSAR's superior capability in dynamically adjusting token allocation based on problem difficulty.

- *Ablation of reasoning window:* As $t_{min}$ is progressively reduced to 0, the robustness of our method declines. Intuitively, early exits may occur at suboptimal points due to inflated

confidence values. Correspondingly, we observe a consistent decrease in both token count and accuracy.

### 4.4 LATENCY ANALYSIS

We evaluate inference latency following the methodology of PMPD (Chen et al., 2025a), reporting *per-task latency* (average seconds per question) to enable direct comparison across methods. Unlike end-to-end dataset latency measurements common in some works, per-task latency isolates hardware efficiency from dataset size variations.

Table 3: Per-task latency (seconds) and token counts across different datasets for DeepSeek-R1-Distill-Qwen-7B.

| Method | GSM8K Lat. | GSM8K Tok. | MATH-500 Lat. | MATH-500 Tok. | AIME120 Lat. | AIME120 Tok. | GPQA Lat. | GPQA Tok. | AMC23 Lat. | AMC23 Tok. |
|---|---|---|---|---|---|---|---|---|---|---|
| 16-bit | 21.3 | 1847 | 40.1 | 4187 | 146.7 | 14910 | 91.6 | 9850 | 67.8 | 6855 |
| 4-bit | 13.3 | 1822 | 25.0 | 4460 | 89.8 | 14141 | 46.2 | 8769 | 42.4 | 6660 |
| 3-bit | 9.6 | 1858 | 18.1 | 4223 | 61.4 | 12481 | 33.8 | 8123 | 30.7 | 7929 |
| **TSAR** | **6.7** | **1354** | **13.6** | **2994** | **48.7** | **9900** | **15.6** | **3762** | **22.4** | **4643** |

The results are shown in Table 3. While static quantization at 4-bit and 3-bit precision is expectedly to reduce latency compared to the 16-bit original, our TSAR framework demonstrates substantially greater gains. Specifically, TSAR achieves speedups ranging from 2.9× to 5.9× across various datasets. Moreover, it attains higher accuracy (as shown in Table 1) than the Uniform 3-bit model while simultaneously achieving lower latency, indicating that TSAR realizes an optimal trade-off between performance and efficiency.

## 5 RELATED WORK

**CoT Reasoning Efficiency.** Recent efforts to improve CoT reasoning efficiency focus on token reduction by shortening CoT paths, building smaller models, or accelerating decoding (Feng et al., 2025; Hashemi et al., 2025; Chen et al., 2024; Lee et al., 2025). Methods to shorten CoT chains include training-dependent approaches like reinforcement learning (RL) with length penalties (Ma et al., 2025b; Li et al., 2025; Aggarwal & Welleck, 2025; Xia et al., 2025; Hou et al., 2025) (e.g., O1-Pruner (Luo et al., 2025), DAST (Shen et al., 2025a)) and supervised fine-tuning (SFT) on variable-length data (Xia et al., 2025; Ma et al., 2025b). Training-free alternatives use prompting to enforce brevity or route queries to specialized models (Renze & Guven, 2024; Ong et al., 2024). Other strategies build smaller, more capable models via knowledge distillation (Feng et al., 2024; Chen et al., 2025b; Shen et al., 2025b) or accelerate decoding with techniques like problem decomposition (Teng et al., 2025) and speculative decoding (Pan et al., 2025). These approaches often require training or treating redundancy dimensions independently, overlooking phase-specific patterns. Unlike them, our TSAR framework provides training-free, phase-aware joint optimization of temporal and spatial redundancy.

**LLM Efficiency.** Model compression is pivotal for alleviating the high resource demands of LLMs, with key strategies including knowledge distillation (Gou et al., 2021), pruning (Frantar & Alistarh, 2023), and quantization (Lin et al., 2024; Frantar et al., 2022). Among these, post-training quantization (PTQ) provides a practical option by compressing models after training while retaining most performance with low overhead. Recent PTQ innovations include SmoothQuant (Xiao et al., 2023), which handles activation outliers for smoother low-bit conversion; GPTQ (Frantar et al., 2022), which fine-tunes quantization layer by layer; and AWQ (Lin et al., 2024), which prioritizes salient weights to maintain generation quality. Versatile systems like Any-Precision LLM (Park et al., 2024) allow runtime selection of bit-widths from a single model without added storage costs, and Progressive Mixed-Precision Decoding (Chen et al., 2025a) varies precision adaptively throughout the decoding sequence. Despite their effectiveness in curbing spatial overhead, these methods typically enforce broad, non-specialized policies that disregard the varying demands of CoT phases.

## 6 CONCLUSION

We presented TSAR, a training-free framework for optimizing Chain-of-Thought reasoning through coordinated temporal and spatial redundancy exploitation. Our approach achieves up to 12.4× speedup while maintaining accuracy without requiring training. The results establish coordinated multi-dimensional optimization as substantially superior to conventional strategies, opening new directions for practical reasoning optimization.

ETHICS STATEMENT

Our work focuses on enhancing the computational efficiency of LLM reasoning using public models and benchmarks. By dynamically adjusting computational precision and stopping criteria without altering model parameters, our method introduces no new ethical risks and inherits the profile of the base models. This approach aims to democratize access to advanced AI for resource-constrained environments and reduce the environmental footprint of LLM deployments.

REPRODUCIBILITY STATEMENT

Our experiments are conducted on public models and benchmarks. For all reported results, we average four runs with different random seeds; the main experiments use a fixed seed (42) for direct replication. We will open-source our complete implementation to ensure full reproducibility and facilitate future research.

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

Appendix

## A THE USE OF LARGE LANGUAGE MODELS (LLMS)

During the writing process, we utilized LLMs, specifically GPT-5, to refine the manuscript's language for clarity and fluency. The authors retained full responsibility for all content, with the LLMs serving exclusively as a tool for language enhancement.

## B JUSTIFICATION FOR EQUATION (5)

The linear structure of our dynamic precision scheduler in Equation 5 is grounded in a first-order Taylor approximation of an optimal, yet intractable, precision function.

### B.1 THEORETICAL FOUNDATION

Let us assume there exists an ideal precision function $p^* = f(t, C)$ that perfectly maps the reasoning progress (time step $t$) and the model's current confidence ($C$) to the optimal number of bits. This function is complex and unknown. To create a practical scheduler, we can approximate it using a first-order Taylor expansion around a specific operational point $(t_0, C_0)$:

$$p^* \approx f(t_0, C_0) + \left.\frac{\partial f}{\partial t}\right|_{(t_0, C_0)} (t - t_0) + \left.\frac{\partial f}{\partial C}\right|_{(t_0, C_0)} (C - C_0) \tag{9}$$

We define our operational point at the start of a reasoning phase, where progress $t_0 = 0$. For the initial confidence $C_0$, we make a foundational definition. At $t = 0$, the reasoning process has not yet begun, meaning no computational steps have been taken and, therefore, no uncertainty or error has been introduced. The concept of "measured confidence" is not yet applicable. We thus define the confidence at this boundary condition to be $C_0 = 1$, representing an idealized state of zero uncertainty before the first computational step.

### B.2 MAPPING TO THE TSAR SCHEDULER

Our proposed scheduler is a direct, practical implementation of Equation 9. We map each term as follows:

- **Zeroth-Order Term:** The value of the function at the operational point, $f(0, 1)$, represents the optimal precision at the beginning of a phase with perfect confidence. This is naturally modeled by our phase-specific baseline:

$$f(0, 1) \triangleq p_{base}(\phi_t) \tag{10}$$

- **First-Order Term (Progress):** The term $\left.\frac{\partial f}{\partial t}\right|_{(0,1)} \cdot t$ captures how precision should change with time. Based on our empirical observations, initial reasoning steps are more critical, implying that the required precision decreases as reasoning progresses. We therefore model the partial derivative as a negative constant:

$$\left.\frac{\partial f}{\partial t}\right|_{(0,1)} \triangleq -\frac{\alpha}{T_{exp}} \tag{11}$$

This leads to the term $-\alpha \cdot \frac{t}{T_{exp}}$ in our scheduler.

- **First-Order Term (Confidence):** The term $\left.\frac{\partial f}{\partial C}\right|_{(0,1)} \cdot (C - 1)$ captures how precision should adapt to changes in confidence. Intuitively, a drop in confidence requires an increase in precision to mitigate potential errors. This implies an inverse relationship, so we model this partial derivative as a different negative constant:

$$\left.\frac{\partial f}{\partial C}\right|_{(0,1)} \triangleq -\beta \tag{12}$$

Substituting this into the Taylor expansion gives the term $-\beta(C-1)$, which is algebraically equivalent to $+\beta(1 - C)$ as used in our scheduler.

By combining these components, our scheduler from Equation 5 emerges as a direct implementation of the first-order approximation:

$$p_t \approx p_{base}(\phi_t) - \alpha \cdot \frac{t}{T_{exp}} + \beta \cdot (1 - C(t)) \tag{13}$$

This principled approach provides a justification for the linear form of our scheduler, demonstrating that it is a well-founded design choice balancing accuracy and computational efficiency.

---

**Algorithm 1** TSAR: Temporal-Spatial Adaptive Reasoning

---

**Require:** Reasoning LLM $M$, $\mathbf{x}$, thresholds $\{\tau_u, \tau_e\}$, weights $\{\alpha, \beta\}$
**Ensure:** Generate sequence $\mathbf{s}$, precision $\mathbf{p}$
1: $p_0 \leftarrow 4, t \leftarrow 0, N_{adapt} \leftarrow 0, H_{conf} \leftarrow []$
2: **while** $t < T_{max}$ **and not** terminated **do**
3:    $s_{t+1} \leftarrow M(s_{1:t}, p_t)$
4:    $H_{conf}$.append($C_{base}(s_{t+1})$)
5:    **if** $\mathcal{S}(s_{t+1})$ **then**
6:       $\phi_{t+1} \leftarrow \Phi(s_{t+1})$
7:       $H_t \leftarrow$ ComputeEntropy($s_{t+1}$)
8:       $S_t \leftarrow$ AssessStability($H_{conf}$)
9:       $C_t \leftarrow$ UnifiedConfidence($\phi_{t+1}, H_t, S_t$)
10:     $prog \leftarrow t/T_{exp}$
11:     $p_{target} \leftarrow \max(p_{base}(\phi_{t+1}) - \alpha \cdot prog - \beta \cdot (1 - C_t) - \gamma \cdot U(\phi_{t+1}), 2)$
12:     **if** $C_t \geq \tau_p$ **and** $p_t > p_{target}$ **then**
13:        $p_{t+1} \leftarrow p_{target}, N_{adapt} \leftarrow N_{adapt} + 1$
14:        ExecuteAdaptation($\Delta t$)
15:     **else**
16:        $p_{t+1} \leftarrow p_t$
17:     **end if**
18:     **if** $C_t \geq \tau_e$ **and** $S_t$ **and** $\mathcal{PC}(\phi_{t+1})$ **and** $t \geq t_{min}$ **then**
19:        InjectCompletion($\phi_{t+1}$)
20:        **return** $\mathbf{s}_{1:t+1}, \mathbf{p}_{1:t+1}$
21:     **end if**
22:    **else**
23:       $p_{t+1} \leftarrow p_t$
24:    **end if**
25:    $t \leftarrow t + 1$
26: **end while**
27: **return** $\mathbf{s}_{1:t}, \mathbf{p}_{1:t}$

---

## C   TSAR ALGORITHM

Algorithm 1 presents our TSAR framework. The algorithm operates by monitoring reasoning generation for thought transition points, which serve as coordination opportunities. At each transition, it performs phase classification, computes unified confidence metrics, and makes coordinated decisions about precision adaptation and potential termination.

## D   DETAILS OF REASONING PHRASE CLASSIFICATION

To understand the internal mechanics of TSAR, we analyze two key components: the accuracy of our phase classifier and the resulting dynamic precision allocation strategy. The effectiveness of our framework hinges on correctly identifying the current reasoning phase to apply the appropriate optimization policy. As shown in Figure 1, our lightweight, keyword-based phase classifier (Eq. (5)) achieves nearly 90% average accuracy (details of this classifier can be found in Table 1). While this keyword-based classifier demonstrates high accuracy and efficiency for the tasks evaluated, we acknowledge that its robustness may vary on out-of-domain problems. Future work could explore

Table 1: Keywords used in reasoning phrase classification.

| Phase | Keywords |
|---|---|
| PF | recall, define, since, condition, theorem, inequality |
| SC | compute, calculate, formula, integral, $\int$, =, +, -, *, /, $\sqrt{}$, sin, cos, [ |
| SV | check, verify, inconsistent, actually, hold on, error, doubt, ?, how, whether, confirm, correct |

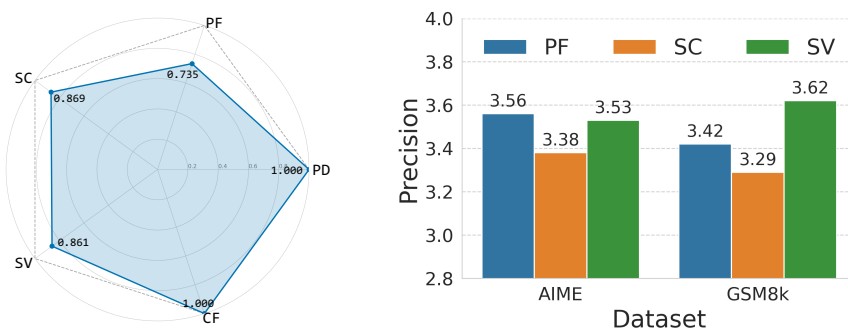

Figure 1: Phrase classification accuracy. Figure 2: Task-adaptive precision allocation.

replacing this with a small, lightweight learned classifier to enhance generalizability without significantly increasing computational overhead.

Building on this accurate phase detection, TSAR demonstrates a sophisticated, task-aware approach to resource management, as illustrated in Figure 2. The framework learns to dynamically allocate precision based on the complexity and nature of the dataset. For the highly complex AIME dataset, the highest precision is allocated to Problem Formulation (PF) (3.56 bits), emphasizing the need to correctly understand and set up the problem. In contrast, for the arithmetically-focused GSM8k dataset, the highest precision is shifted to Solution Verification (SV) (3.62 bits), reflecting the critical importance of rigorously checking the final computed answer. This adaptive behavior confirms that TSAR does not use a one-size-fits-all policy; instead, it intelligently distributes computational resources to the reasoning phases where they are most impactful, tailoring its strategy to the unique demands of each task.

## E COMPLETION INDICATORS

To implement natural termination, we employ confidence-based completion phrase injection. The contextually appropriate completion indicators employed in our experiments are as follows,

- **Prompt 1: `Final Answer`** The usage of this prompt stems from our empirical observation of model outputs. We consistently observed that the model generates the token sequence `Final Answer` immediately preceding its final output. We therefore hypothesize that this prompt effectively triggers early exit behavior. Experimental results demonstrate that while this prompt achieves excellent truncation performance, it inadvertently suppresses the generation of the solution reasoning component, thereby exerting non-negligible negative impacts on final accuracy.

- **Prompt 2: `Okay, I think I have finished thinking.`** This formulation draws inspiration from Ma et al. (2025a), where the original work employed it at the beginning of model outputs to skip chain-of-thought reasoning. We posit that inserting this prompt within reasoning chains can effectively induce early exit. Our experiments reveal that this prompt maintains an optimal balance between truncation efficiency and solution reasoning length, consequently enabling the model to simultaneously optimize both token generation quantity and prediction accuracy.

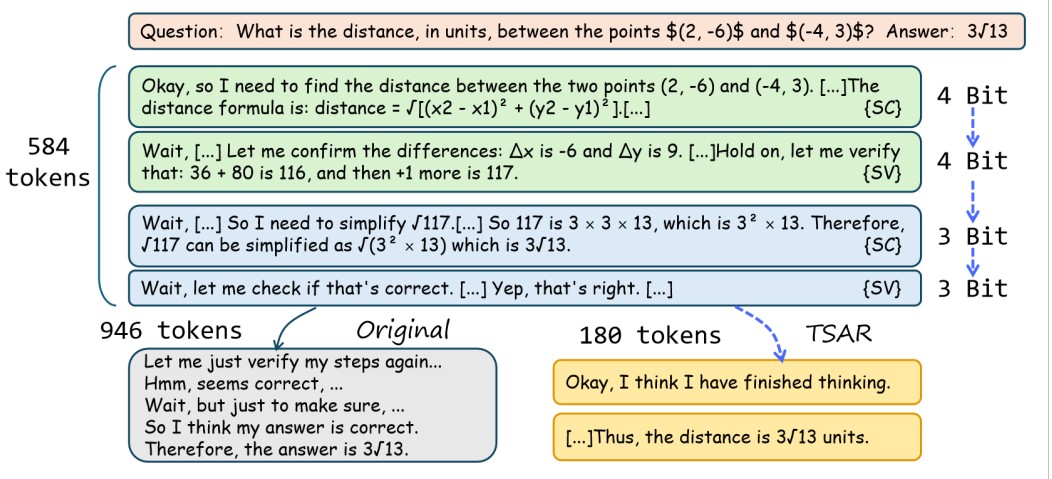

Figure 3: A case study demonstrating TSAR's optimization process. TSAR identifies reasoning phases, adaptively reduces precision for computational steps, and terminates early upon reaching a stable conclusion, pruning the redundant verification steps.

# F  CASE STUDY

To provide a granular view of our framework's mechanics, we present a case study on a challenging problem from the MATH-500 dataset. This analysis contrasts the lengthy, resource-intensive reasoning process of the original DeepSeek-R1-Distill-Qwen-7B model with the highly efficient, adaptive process guided by our TSAR framework. The comparison, illustrated in Figure 3, reveals how TSAR dynamically prunes both spatial and temporal redundancy without compromising the final answer's accuracy.

It can be observed that after generating 584 tokens, the original model continues to produce an additional 946 tokens. In contrast, when applying our TSAR (Token-Scalable Adaptive Reasoning) method, the bit allocation process (indicated by blue arrows) can be observed, and the corresponding text output is nearly identical to that of the original model. Upon detecting that the reasoning quality meets the coordinated early-termination criterion, TSAR inserts the phrase "Okay, I think I have finished thinking." to halt further generation by the original model. Subsequently, the quantized model with TSAR generates only 180 tokens before concluding the reasoning process.

Although both approaches ultimately produce correct answers, our TSAR method achieves dual improvements in temporal efficiency (reducing inference time) and spatial efficiency (optimizing computational resource usage).

