# OpenReview forum: "Optimizing Temporal and Spatial Efficiency for Chain-of-Thought Reasoning in Large Language Models"
_ICLR.cc/2026/Conference — ICLR 2026 Conference Withdrawn Submission_

### Official Review · Reviewer_5DDz · 2025-10-28

**Soundness:** 2
**Presentation:** 1
**Contribution:** 2
**Rating:** 2
**Confidence:** 3

**Summary:**

This paper proposes Spatio-Temporal Adaptive Reasoning (TSAR), a training-free framework that enhances the efficiency of CoT reasoning by leveraging temporal and spatial redundancies. While the paper's motivation is timely and relevant, it suffers from conceptual ambiguity, methodological opacity, a limited theoretical foundation, and overstated contributions.

**Strengths:**

- This work addresses the significant problem of reducing the inference cost of Chain-of-Thought (CoT) reasoning.
- This work proposes a training-free optimization perspective.
- This work claims strong empirical results, albeit unconvincing.

**Weaknesses:**

- **Lack of Clarity in Core Mechanisms:** Key figures and equations are either missing or uninformative, which significantly limits reader comprehension. Crucially, the paper fails to describe the formulas for core computations such as Confidence and Stability. The calculation methods for key parameters, specifically **$r_{\phi}(s_t)$** and **$H_{\text{base}}$**, are not explained, leaving their implementation entirely unclear. The paper overuses high-level conceptual phrases without grounding them in low-level implementation details or pseudocode. Consequently, and due to the absence of crucial experimental details, the results are difficult to **reproduce or verify**. Even though providing supplementary material, it is still disorganized and fails to get the necessary implementation specifics.
- **Unfair Experimental Comparisons:** The performance claims should be benchmarked against established quantization models, such as "one-bit quantization" methods.  For a fair comparison, the baselines (e.g., s1) must also undergo the same quantization during inference. The current comparison is inequitable. Moreover, baselines like s1 can operate under a fixed token budget. If the experimental settings for budget control are inconsistent between the proposed method and the baselines, the comparison is rendered meaningless.
- **Lack of Analysis:** The "progressive precision reduction" strategy lacks clear visualization. A more in-depth case study is necessary to elucidate how this process is implemented in practice. The paper lacks critical ablation studies on the contribution of keywords or the impact of the priors, making it difficult to assess the individual components of the framework.
- **Limited Generalizability and Robustness:** The method appears heavily reliant on pre-computed statistics and static reasoning patterns of specific models. Reasoning patterns are known to vary significantly across different model families (e.g., Claude vs. Grok), which raises serious concerns about the method's generalizability and portability. It is unclear whether different benchmarks require distinct prior thresholds. If so, the reported performance gains may not be genuinely innovative but rather an artifact of task-specific, manual tuning or "cherry-picking".
- **Missing Related Work:**

[1] Towards reasoning era: A survey of long chain-of-thought for reasoning large language models

[2] From System 1 to System 2: A Survey of Reasoning Large Language Models

[3] Wait, We Don't Need to" Wait"! Removing Thinking Tokens Improves Reasoning Efficiency

[4] Aware First, Think Less: Dynamic Boundary Self-Awareness Drives Extreme Reasoning Efficiency in Large Language Models

[5] AdaCtrl: Towards Adaptive and Controllable Reasoning via Difficulty-Aware Budgeting

**Questions:**

- Could you re-clarify the core mechanisms of your method? The paper currently lacks the specific formulas for core computations like Confidence and Stability.
- How are key parameters such as $r_{\phi}(s_t)$ and $H_{\text{base}}$ calculated?
- To ensure a fair comparison, why wasn't the method benchmarked against established quantization models?
- Why not consider to compare all baselines with the same quantization during inference?
- How was the token budget controlled consistently between your method and the baselines? Without consistent settings, the comparison seems inequitable and potentially meaningless.
- Could you provide a more in-depth analysis, like a visualization or case study would greatly clarify the "progressive precision reduction" strategy.
- Have you performed ablation studies to isolate the impact of keywords/behaviors and the priors? Understanding the contribution of each reasoning behavior is crucial for evaluating the method's design.
- How generalizable is your method across different model families (e.g., Claude vs. Grok), given its reliance on pre-computed statistics and static reasoning patterns?
- Do different benchmarks require manually tuning distinct prior thresholds? If so, how can we be sure the performance gains are not simply an artifact of task-specific "cherry-picking" rather than a genuinely innovative contribution?
- Adding more reference

I am willing to revise my score if these questions are addressed and clarified in a revision.

---

### Official Review · Reviewer_r8Rq · 2025-10-29

**Soundness:** 2
**Presentation:** 2
**Contribution:** 2
**Rating:** 4
**Confidence:** 3

**Summary:**

This paper aims to improve the efficiency of chain of thought in LLMs by a strategy that combines adaptive quantization and a better stopping criterion. The main idea is to partition the CoT into phases, and adjust the precision individually for each phase, via a linear function. The termination criterion goes by a confidence, which combines a relative uncertainty with a quality measure. The strategy is not learned (or maybe the linear function is learned?), and fairly light-weight when combined with recent dynamic precision techniques. It is evaluated on a set of reasoning tasks, and compared to state of the art methods for quantization and termination.

**Strengths:**

- The proposed method is simple and lightweight, and should be relatively easy to apply.

- The empirical results in Table 1 for two larger LLMs (7B, 8B parameters) are good (fewer tokens, less precision, decent accuracy), and the method seems to outperform purely quantization and purely termination rules.

- The division of the reasoning process into phases and an adaptive mechanism for each phase makes intuitive sense and seems to be effective, too.

**Weaknesses:**

- It is not fully clear where exactly most of the gains are coming from. For instance, PMPD (Cheng et al, 2025a) is also developing an adaptive quantization framework, but without the termination rule, if I understand correctly. However, the proposed method seems to be more "efficient" than PMPD. Is the gain coming from the earlier termination or from a slightly different quantization schedule? This is not explored; the ablations only include parameters of the termination criterion.

- Novelty: the idea of the paper is interesting. Yet, PMPD has a somewhat similar motivation and adaptive quantization strategy: they also partition the reasoning sequence into phases (just different ones), and they also decrease precision towards the end. What exactly is the new insight / strategy of the quantization schedule? How does it differ? The derivation of the linear function is quite simple; is the Taylor series / linearization in any way specific to this problem?
In what way is the termination criterion different from prior work?

- Clarity: several concepts and notations are not defined at all, or not defined when they are introduced. Some procedures are also not specified. A few examples:
* when I first read the paper, I only understood towards the middle/end of the paper that "temporal redundancy" simply means that the model goes on after it has essentially found the answer. I was thinking of unnecessary "fluff" during the reasoning process, too, and was wondering how to identify it on the fly.
* how exactly is "efficiency" in Table 1 defined? As a formula?
* what is $\phi_t$ (eqn (5)),$ \phi(s_t)$ , $r_{\phi(s_t)}$ (eqn (7))?
* how are the parameters in eqn 5 selected? And the parameters and constants in eqns 7 and 8?
* see also my questions below

**Questions:**

- Sec. 2: what model was used for these experiments?

- line 150/151: "to statistically analyze the scores across these stages". The Qwen reward model seems to be scoring entire responses. How exactly did you score specific phases, i.e., assign a reward to each phase?

- The keywords in equations 3 and 4 are hand-selected. How about learning a simple linear classifier? How would you select the keywords that characterize different phases?

- Eqn 5: How are $p_{base}$, $\alpha$, $\beta$, $T_{exp}$ determined? Are they specific to an LLM model or applicable across different models?

- what is $\phi_t$ (eqn (5)),$ \phi(s_t)$ , $r_{\phi(s_t)}$ (eqn (7))? How is $r_{\phi(s_t)}$ computed?

- Sec 3.4: what is the reasoning behind selecting only transition vocabulary to estimate uncertainty?

- Sec 4.2 (Main results): how exactly is "efficiency" defined, as a formula?
The table caption says it combines spatial and temporal efficiency. What exactly does this mean?

- How are the parameters and constants in eqns 7 and 8 selected?

---

### Official Review · Reviewer_b83C · 2025-10-31

**Soundness:** 2
**Presentation:** 2
**Contribution:** 2
**Rating:** 4
**Confidence:** 4

**Summary:**

The paper introduces a new training-free framework TSAR to solve two dimensions of the redundancy: temporal and spatial. Authors provides a classification method using the Dewey’s model to describe the sequential process. Enough performance evaluation was done on several mathematical and general reasoning benchmark(e.g., GSM8K, MATH-500, AIME-120, MMLU) for two types of models (DeepSeek-R1-Distill-Qwen-7B and Qwen-3-7B) which shows up to 5.3x-12.4x inference speedup for special datasets. Authors also optimize redundancy from the coordinate terminations and phase-aware quantization aspect with the ablation analysis.

**Strengths:**

1.The two-dimension redundancy optimization perspective is novel with the clear motivations and observations. The training-free framework also important for the CoT reasoning redundancy improvement.

2.Comprehensive evaluation shows the efficiency improvements with maintaining accuracy. The selection of datasets can also include different fields.

**Weaknesses:**

1.The two-dimension redundancy optimization perspective is novel with the clear motivations and observations. The training-free framework also important for the CoT reasoning redundancy improvement.

2.Comprehensive evaluation shows the efficiency improvements with maintaining accuracy. The selection of datasets can also include different fields.

**Questions:**

1.In the paper, both models have a scale of 7B, but in fact, LLMs of different scales have significant differences in reasoning ability, behavioral patterns, and redundancy performance. Do you think the effectiveness of TSAR, especially its keyword based stage classification and entropy driven termination strategy, can be directly extended to larger scale models?

2.What do you think is the reason why the GPQA Diamond MC Dataset is significantly more efficient than other datasets? Is it because the combination of datasets and model classification methods are more compatible?

3.I am concerned about the value of the Dewey model, perhaps it is necessary to compare different classification methods? For example, I only used 3 categories(question-solving-reflection) instead of 5 categories.

4.Do the three challenges have corresponding references support?

5.For the keyword mechanism, how does the system handle conflicts if a fragment contains keywords from multiple stages simultaneously? Is there a priority rule, or how does this ambiguity affect subsequent accuracy allocation and termination decisions?

6.Whether the termination criteria are reliable and whether different models can effectively judge without training if the completion indicators are different.

7.For Fig5, How is this T_exp estimated at the beginning of the inference? Is it a fixed hyperparameter at the dataset level, or a dynamically predicted value for each input?

8.For the framework you proposed, if a critical computational step (such as a complex multiplication) occurs in the middle of a long segment that is judged as having low precision requirements overall, will this introduce errors?

9. Section 3.3 mentions that the precision scheduler is a linear model. However, the inference process of LLM and its sensitivity to accuracy are highly likely to be highly nonlinear. Is the choice of linear model mainly based on considerations of computational efficiency, or is there stronger theoretical or empirical evidence that it is sufficient to describe this relationship? Have you tried simpler nonlinear functions?

10.I am curious about the reasons for the gain of datasets related to mathematics and fundamental knowledge. Dewey's five stage cognitive model is actually more in line with mathematics and logical reasoning. But for unstructured reasoning tasks such as history, law, or philosophy in MMLU, will TSAR's performance gain in the latter only be due to the effective early stopping of short answers?

---

### Official Review · Reviewer_tC8f · 2025-11-12

**Soundness:** 2
**Presentation:** 2
**Contribution:** 3
**Rating:** 6
**Confidence:** 3

**Summary:**

The paper addresses the computational inefficiency of CoT reasoning in LLMs caused by temporal and spatial redundancy. It proposes Temporal-Spatial Adaptive Reasoning (TSAR), a training-free framework that segments reasoning via Dewey’s reflective thinking model, applies progressive precision reduction, and coordinates termination with entropy-based confidence estimation. Experiments on 6 datasets and 2 models show TSAR achieves up to 12.4× speedup while maintaining accuracy, outperforming static quantization and early termination baselines.

**Strengths:**

* The multi-dimensional redundancy exploitation is innovative and well-grounded. The framework jointly optimizes temporal (unnecessary reasoning steps) and spatial (excessive precision) redundancy, which are ignored by existing methods.
* The experimental evaluation is comprehensive. Evaluations cover multiple mathematical and general-purpose tasks, with ablation studies confirming the contribution of key components.

**Weaknesses:**

* The method’s applicability to larger or smaller models is untested: experiments only use 7B and 8B models, while LLMs with different scales of parameters may exhibit different redundancy patterns. If the time is limited, the authors can provide some results on smaller models like DeepSeek-R1-Distill-Qwen-1.5B.
* The keyword-based phase classifier may lack generalizability and it relies on domain-specific keywords, although the paper acknowledges potential fragility on out-of-domain tasks. Additionally, how can the authors ensure that the model's response **could be" classifed into the five distinct phases? How does TSAR handle reasoning sequences with ambiguous phase transitions (i.e., lacking predefined keywords), and what is the accuracy of phase classification in such cases?

**Questions:**

See weaknesses. Also:
* In resource-constrained edge devices with strict memory limits, does the Any-Precision quantization’s overhead offset TSAR’s efficiency gains?
* For tasks with highly variable reasoning chain lengths (e.g., open-ended creative reasoning or code generation tasks), how does the termination threshold affect the balance between accuracy and efficiency, and is there an adaptive threshold adjustment mechanism?

---

### Note · Authors · 2026-01-06

I have read and agree with the venue's withdrawal policy on behalf of myself and my co-authors.